# Spectral and Radiometric Measurement Requirements for Inland, Coastal and Reef Waters

**Peter Gege [1],[*]**  **and Arnold G. Dekker [2]**

1    Deutsches Zentrum für Luft- und Raumfahrt (DLR), Earth Observation Center, Remote Sensing Technology Institute, Oberpfaffenhofen, 82234 Wessling, Germany

2    CSIRO, Oceans & Atmosphere, GPO Box 1700, Canberra ACT 2601, Australia; arnold.dekker@csiro.au

*    Correspondence: peter.gege@dlr.de; Tel.: +49-8153-281242

**Abstract:** This paper studies the measurement requirements of spectral resolution and radiometric sensitivity to enable the quantitative determination of water constituents and benthic parameters for the majority of optically deep and optically shallow waters on Earth. The spectral and radiometric variability is investigated by simulating remote sensing reflectance ($R_{rs}$) spectra of optically deep water for twelve inland water scenarios representing typical and extreme concentration ranges of phytoplankton, colored dissolved organic matter and non-algal particles. For optically shallow waters, $R_{rs}$ changes induced by variable water depth are simulated for fourteen bottom substrate types, from lakes to coastal waters and coral reefs. The required radiometric sensitivity is derived for the conditions that the spectral shape of $R_{rs}$ should be resolvable with a quantization of 100 levels and that measurable reflection differences at at least one wavelength must occur at concentration changes in water constituents of 10% and depth differences of 20 cm. These simulations are also used to derive the optimal spectral resolution and the most sensitive wavelengths. Finally, the $R_{rs}$ spectra and their changes are converted to radiances and radiance differences in order to derive sensor (noise-equivalent radiance) and measurement requirements (signal-to-noise ratio) at the water surface and at the top of the atmosphere for a range of solar zenith angles.

**Keywords:** remote sensing; inland water; phytoplankton; colored dissolved organic matter (CDOM); non-algal particles; shallow water; coral reef; reflectance; spectral resolution; radiometric sensitivity

## 1. Introduction

Many Earth-observing satellite sensors have been developed with the primary objectives of serving either terrestrial, oceanic or atmospheric remote sensing applications [1]). Since no satellite sensor specifically designed for coastal and inland waters is available at present [2–4], data from these sensors are used for freshwater, estuarine and coastal water quality observations, bathymetry and benthic mapping [5–7]. However, such land and ocean specific sensors are not designed for these complex aquatic environments and consequently are not likely to perform as well as a specialized coastal and inland water mapping sensor would.

Due to a lack of comprehensive information that would allow the design of a spaceborne sensor system optimized for the global monitoring of inland and coastal waters, the Committee on Earth Observation Satellites (CEOS) initiated a study [1] which collected literature information and conducted simulations to derive requirements that could form the basis for developing such a dedicated sensor system. It identified the following parameters that such a system should be able to quantify: (i) algal pigment concentrations of chlorophyll-a, accessory pigments relevant for phytoplankton functional types research, phycoerythrin and phycocyanin for monitoring cyanobacteria; (ii) algal fluorescence, especially chlorophyll-a fluorescence at 684 nm; (iii) suspended matter concentration, possibly split up

into organic and mineral components; (iv) absorption coefficient of colored dissolved organic matter (CDOM); (v) spectral slope of CDOM absorption to discriminate terrestrial from marine CDOM; (vi) spectral absorption and backscattering coefficients of the optically active components; (vii) measures of water transparency such as Secchi disk depth, vertical attenuation coefficient and turbidity index. For optically shallow waters, the following parameters were also identified: (viii) water column depth to derive bathymetry; (ix) substratum type and cover, such as muds, sands, coral rubble, seagrasses, macro-algae and corals; (x) plants floating at or just above the water surface.

The accuracy with which these parameters can be determined depends on a number of environmental and measurement conditions. The CEOS study [1] addressed these conditions for satellite observation and discussed the related mission and sensor concepts. There is a trade-off in spectral, spatial and radiometric resolution as the number of photons available to be measured will constrain choices. For example, detecting a change of 10% in chlorophyll from a low Earth orbiter (LEO) satellite sensor at a high latitude in a northern or southern hemisphere winter requires a totally different radiometric sensitivity than detecting a similar change in the tropics with the sun close to zenith. One may think this can be solved by a geostationary orbit, but then the look angle from the equatorial position to the high latitudes becomes a limiting factor. In addition, spatial resolution and integration time also dictate how many photons are available to be measured: the finer spatial resolution advised for e.g., benthic feature mapping or smaller inland waters leads to reduced signal-to-noise ratios (SNRs).

To obtain an overview of the environmental and measurement conditions that are relevant for the majority of inland and coastal waters on Earth, a sensitivity analysis was made for the CEOS report (Appendix A.2 in [1]). The simulations covered optically deep and shallow waters for concentrations of water constituents and for substratum cover types that are representative of many inland, coastal and shallow benthic environments across the world. The studied scenarios represent inland, estuarine, deltaic and near coastal waters with a variety of bottom substrates, such as sand, rock, silt, macrophytes, macro-algae, sea grasses and corals in the shallow areas. The derived set of parameters included the most relevant wavelengths and optimal spectral resolutions for capturing the spectral shape of remote sensing reflectance ($R_{rs}$), wavelengths of maximum sensitivity, noise-equivalent $R_{rs}$ differences ($\Delta R_{rs}$) required to resolve 10% concentration changes in water constituents, and for selected dark and bright water types the corresponding radiances ($L$), noise-equivalent radiance differences ($\Delta L$) and SNRs at the top of the atmosphere for sun zenith angles of 10° and 70° and horizontal visibilities of 10 and 80 km. To support a multispectral sensor design, a wavelength table extending the International Ocean-Colour Coordinating Group IOCCG recommendations for ocean color sensors [8] was derived.

Since the CEOS report was not subject to peer review and the simulated spectra are not publicly available, the authors were repeatedly asked for a citable documentation of the methods and results of the sensitivity analysis and for access to the software and the complete simulation dataset. For this reason, we repeated all simulations and statistical analyses with slightly updated settings for the scenarios, improved approaches for simulating radiometric sensor ($\Delta L$) and measurement (SNR) requirements and extended statistical analysis of the simulated spectra. A method was developed for parameterizing the SNR in terms of $R_{rs}$, $\Delta R_{rs}$ and atmospheric path radiance and for separating the SNR contributions from the ground and the atmosphere. This parameterization, which, to our knowledge, has not been described in the literature before, allowed us to specify the measurement requirements more systematically compared to the CEOS report. The updated scenarios, methods and simulations, their statistical analysis and the derived measurement and sensor requirements are described in this paper. The software used to perform the simulations and the full dataset of simulated spectra are available online [9] (Supplementary Materials, https://doi.org/10.5281/zenodo.3817616).

This paper focuses on an aspect which, as far as we know, has not been performed systematically before: to approach the required reflectance and radiance sensitivities using thorough physics-based simulation for actually measured concentration ranges and optical properties from representative water bodies, as well as actual measured benthic substratum spectra. The described methods and results are

generic in nature and thus may provide sensor engineers and scientists with information to design Earth observation sensors and juxtapose what is technologically feasible with what is scientifically desirable.

## 2. Materials and Methods

The derivation of widely applicable measurement requirements follows the following concept. First, a number of scenarios are defined to represent the variability of the optical properties for typical and extreme lakes on Earth (Section 2.1). The optical variability within each scenario is mainly determined by the concentration changes in phytoplankton, CDOM and non-algal particles, by the variability of the spectral slope of CDOM absorption in the case of optically deep water and by water depth and bottom substratum types in the case of optically shallow water. An analytical model is used for simulating a large number of remote sensing reflectance spectra for different values of the variable parameters within the scenario-specific ranges. This model is described in Section 2.2. The set of simulated spectra is subsequently analyzed to determine the optimal spectral and radiometric resolution of measurements. The methods for deriving these measurement requirements are described in Section 2.3 (optimal spectral resolution) and Section 2.4 (optimal radiometric sensitivity).

### 2.1. Scenarios

The inland and coastal waters on Earth are as variable as their surrounding ecosystems and catchment areas. Water constituents differ considerably in type, concentration and thus optical properties. The various sources of organic and inorganic material make the reflectance spectra more variable than for the open ocean. When the bottom is visible at the surface in shallow waters, the reflectance spectrum is affected by the optical properties of the substratum (the inanimate bottom material), as well as the benthos (the living organisms on the substratum). These waters are called optically shallow waters, as opposed to the optically deep waters, where the radiance or reflectance signal measured at the surface comes from backscattering and fluorescence in the water column. To study the variability of reflectance, we used two scenarios: one representing values of optically active water quality components (OACs) that can be considered as typical for inland and coastal waters and one considering extremely high levels of OACs, representing extreme aquatic ecosystem conditions. Each scenario is defined by a set of measurable quantities, which are related to the spectral variability of reflectance, and by their minimum, maximum and typical values.

Remote sensing groups the OACs into three main classes: phytoplankton, non-algal particles (NAP) and colored dissolved organic matter (CDOM) [2]. NAP consists of varying ratios of organic to mineral matter. Optical properties that are independent of the illumination, such as the absorption coefficient and the backscattering coefficient, are called IOPs (inherent optical properties) or SIOPs (specific inherent optical properties) when normalized to concentration or, in the case of CDOM, normalized to the absorption value of 440 nm. These wavelength-dependent functions, together with the concentrations, allow the simulation of the reflectance of the water body. In the case of optically shallow waters, additionally, the spectral irradiance reflectance (or albedo) of the bottom substrates and benthos are required for simulating the combined reflectance, including the attenuated and backscattered light in the water column.

### 2.1.1. Optically Deep Water

The wavelength-independent parameters defining an optically deep water scenario are the concentrations of phytoplankton (C), NAP (X) and CDOM (Y). Additionally, the spectral slope of CDOM absorption (S) is used to model the natural variability of the spectral shape of CDOM absorption. The wavelength-dependent optical properties are the SIOPs of phytoplankton and NAP. The relationship of these parameters and SIOPs to the reflectance of water is outlined in Section 2.2.

As the concentrations of water constituents are not completely independent from each other, the definition of scenarios is oriented toward certain types of lakes. Scenarios for typical lakes are defined in Table 1, based on actual measured values. They are specified in terms of a typical value and a

representative range of C, X, Y and S. The simulations vary C, X, Y and S across the scenario-specific ranges and keep the SIOPs constant. These results are also valid for most coastal waters, as coastal waters tend to have ranges of concentrations that fall within the inland water ranges.

**Table 1.** Standard scenarios for optically deep water. A scenario is defined by a constant parameter marked as bold. The scenario name consists of the parameter acronym and a sign indicating a low (−) or high (+) value. The variable parameters are specified by a typical value and a range in the notation typical (min-max).

| Scenario | C− | C+ | X− | X+ | Y− | Y+ |
|---|---|---|---|---|---|---|
| Represents | Low phy | High phy | Low NAP | High NAP | Low CDOM | High CDOM |
| Example | Lake Garda | 2 Finnish l. | L. Constance | The Netherlands | L. Maggiore | Lake Peipsi |
| C (mg m$^{-3}$) | **1** | **40** | 2 (0.5–15) | 25 (10–50) | 1 (0.2–5) | 5 (1–20) |
| X (g m$^{-3}$) | 1 (0.2–20) | 10 (5–15) | **1** | **15** | 1 (0.2–10) | 5 (1–10) |
| Y (m$^{-1}$) | 0.1 (0.04–2) | 2.5 (1.5–4) | 0.5 (0.2–2) | 1 (0.5–1.5) | **0.2** | **2.5** |
| S (nm$^{-1}$) | 0.014 (0.01–0.02) | 0.014 (0.01–0.02) | 0.014 (0.01–0.02) | 0.014 (0.01–0.02) | 0.014 (0.01–0.02) | 0.014 (0.01–0.02) |

The concentrations and ranges for scenarios C−, Y− and Y+ are based on Table 1 in [10,11], scenario X+ on lakes in The Netherlands [12] and scenario C+ on the two Finnish lakes, Tuusulanjärvi and Hiidenvesi [13]. S is in most cases between 0.010 nm$^{-1}$ for humic acid dominated waters and 0.020 nm$^{-1}$ when fulvic acids prevail, with a value of 0.014 nm$^{-1}$ being representative of a great variety of water types [14,15].

Of particular relevance for defining measurement requirements are the extreme cases: if a sensor is suitable for the extremes, it should provide even better data in-between. This concept of extremes defines the scenarios of, based on actual measured values (Table 2). The extreme concentrations of C, X and Y are chosen close to a minimum or maximum of Table 1 in [11].

**Table 2.** Extreme scenarios for optically deep water. The notation is the same as in Table 1, except the label −−of a scenario name indicating an extremely low parameter value and ++ an extremely high value.

| Scenario | C−− | C++ | X−− | X++ | Y−− | Y++ |
|---|---|---|---|---|---|---|
| Extreme for | Low phy | High phy | Low NAP | High NAP | Low CDOM | High CDOM |
| Example | Italian lakes | Lake Taihu | Lake Garda | Lake Taihu | Lake Garda | Finnish lakes |
| C (mg m$^{-3}$) | **0.2** | **1000** | 1 (0.1–10) | 20 (1–1000) | 1 (0.1–10) | 5 (1–10) |
| X (g m$^{-3}$) | 1 (0.2–20) | 50 (10–300) | **0.1** | **300** | 1 (0.2–20) | 2 (0.5–5) |
| Y (m$^{-1}$) | 0.1 (0.04–2) | 1 (0.2–3) | 0.1 (0.04–2) | 1 (0.2–3) | **0.04** | **10** |
| S (nm$^{-1}$) | 0.014 (0.01–0.02) | 0.014 (0.01–0.02) | 0.014 (0.01–0.02) | 0.014 (0.01–0.02) | 0.014 (0.01–0.02) | 0.014 (0.01–0.02) |

2.1.2. Optically Shallow Water

The simulations for optically shallow waters alter the bottom substrate or benthos type and water depth, and they keep the parameters of the water column constant. As remote sensing of optically shallow waters favors clear water conditions, the typical concentrations of scenario Y− (C = 1 mg m$^{-3}$, X = 1 g m$^{-3}$, Y = 0.2 m$^{-1}$) are used to specify the water layer. These concentrations are close to the lower end for the standard scenarios; hence, the water is clear for inland waters compared to most other optically deep water scenarios. For coastal waters, these low concentration values are more representative of an average coastal water.

The optical properties of the optically shallow water scenarios are defined by the spectral albedo of the bottom substrates and benthos. Fourteen substrate and benthos types are selected for approximating the natural variability of the water body's bottom reflective properties (Table 3, Figure 1):

**Table 3.** Bottom substrates used for shallow water simulations.

| No. | Substrate Type | Reference |
|:---:|:---:|:---:|
| 0 | Chara contraria (macrophyte) | [16] |
| 1 | Potamogeton perfoliatus (macrophyte) | [16] |
| 2 | Rock | [17] |
| 3 | Bleached coral | [18] |
| 4 | Dark silt | [17] |
| 5 | Bright sand | [18] |
| 6 | Yellow *porites* sp. (coral) | [18] |
| 7 | Purple encrusting coralline algae | [18] |
| 8 | Brown *porites* sp. (coral) | [18] |
| 9 | Posidonia australia (seagrass) | [19] |
| 10 | Detritus (sea-grass wrack) | [17] |
| 11 | Ecklonia radiata (kelp) | [17] |
| 12 | Coarse coral rubble | [18] |
| 13 | Dark sand | [20] |

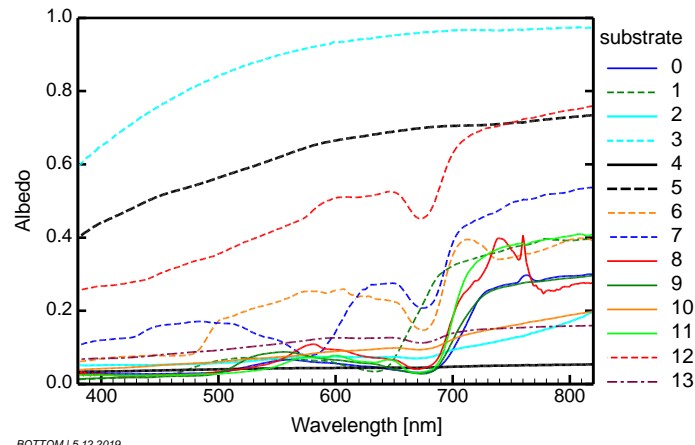

**Figure 1.** Bottom substrate and benthos albedos used for the simulations. See Table 3 for the labeling.

Substrates 0 and 1 were measured in Lake Constance, Germany [16], substrates 2 to 12 in the tropical Lihou Reef National Marine Park and subtropical Lord Howe Island Marine Parks, Australia [17–19] and substrate 13 in the Baltic Sea, Germany [20]. The albedo spectra of these substrates were used as input for the simulations in optically shallow waters.

*2.2. Model*

The reflectance of water depends on the spectral absorption coefficient, $a(\lambda)$, and spectral backscattering coefficient, $b_b(\lambda)$, of the water layer. The most relevant components contributing to $a(\lambda)$ and $b_b(\lambda)$ are pure water (index "W"), phytoplankton (index "phy"), non-algal particles (index "NAP") and CDOM. Their absorption and backscattering coefficients are additive:

$$a(\lambda) = a_w(\lambda) + C \times a_{phy}^*(\lambda) + X \times a_{NAP}^*(\lambda) + Y \times exp\{-S \times (\lambda - 440)\}, \tag{1}$$

$$b_b(\lambda) = b_{b,w}(\lambda) + C \times b_{b,phy}^*(\lambda) + X \times b_{b,NAP}^*(555) \times \left(\frac{\lambda}{555}\right)^{-n}. \tag{2}$$

$C$ is the phytoplankton concentration in units of mg m$^{-3}$ of chlorophyll-a, $X$ is the total suspended matter concentration in units of g m$^{-3}$ and $Y$ is the CDOM absorption at 440 nm in units of m$^{-1}$. While these wavelength-independent parameters are used to model the concentrations of the water constituents, their optical properties are simulated using the wavelength-dependent SIOPs shown

in Figure 2. The specific absorption coefficients of phytoplankton ($a^*_{phy}(\lambda)$) and non-algal particles ($a^*_{NAP}(\lambda)$) and the specific backscattering coefficient of phytoplankton ($b^*_{b,phy}(\lambda)$) are taken from measurements, while CDOM absorption and NAP backscattering are approximated using analytical equations. The parameters of these empirical equations are $S$, the spectral slope of CDOM absorption in units of nm$^{-1}$, $n$, the Angström exponent of NAP backscattering and $b^*_{b,NAP}(555)$, the specific backscattering coefficient of NAP at 555 nm in units of m$^2$ g$^{-1}$.

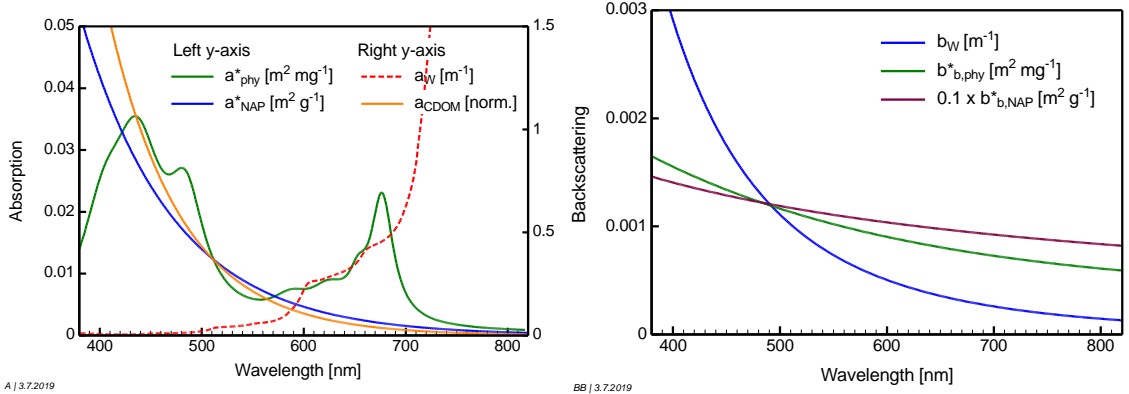

**Figure 2.** Absorption coefficients (left) and backscattering coefficients (right) used for the simulations. The units are given in the legend.

All calculations simulate measurements of remote sensing reflectance $R_{rs}$, which is the ratio of upwelling radiance to downwelling irradiance, both above the water surface and excluding specular reflections at the surface. $R_{rs}$ is related to the corresponding underwater ratio $r_{rs}$ as follows [21,22]:

$$R_{rs}(\lambda) = \frac{\zeta \times r_{rs}(\lambda)}{1 - \Gamma \times r_{rs}(\lambda)}. \tag{3}$$

$\zeta \approx 0.52$ is the water-to-air radiance divergence factor, and the denominator with $\Gamma \approx 1.6$ accounts for the effects of internal reflection from water to air. The model of Albert [23,24] is used for the simulations, which expresses $r_{rs}$ as a polynomial of fourth order of the IOP.

$$u(\lambda) = \frac{b_b(\lambda)}{a(\lambda) + b_b(\lambda)}. \tag{4}$$

The model can be used for optically deep and shallow waters and accounts for the sun zenith angle and the viewing angle. Its coefficients have been derived using Hydrolight [21] simulations, covering wide ranges of environmental parameters, including all concentrations of the standard scenarios and most of the high concentrations of the extreme scenarios. A similar model has been developed by Lee et al. [22,25] for narrower ranges; see [26] for a comparison of the equations and parameter ranges. The following is Albert's equation for optically deep water:

$$
\begin{aligned}
r_{rs}^{deep}(\lambda) = \\
0.0512 \times u(\lambda) \times \left(1 + 4.6659 \times u(\lambda) - 7.8387 \times u(\lambda)^2 + 5.4571 \times u(\lambda)^3\right) \\
\times \left(1 + \frac{0.1098}{\cos \theta'_{sun}}\right) \times \left(1 + \frac{0.4021}{\cos \theta'_v}\right) \times (1 - 0.0044\, v_w);
\end{aligned} \tag{5}
$$

and the following is the equation for optically shallow water:

$$
\begin{aligned}
r_{rs}^{shallow}(\lambda) = r_{rs}^{deep}(\lambda) \times \left[1 - A_{rs,1} \times exp\{-(K_d(\lambda) + k_{uW}(\lambda)) \times z_B\}\right] \\
+ A_{rs,2} \times R_{rs}^b(\lambda) \times exp\{-(K_d(\lambda) + k_{uB}(\lambda)) \times z_B\}.
\end{aligned} \tag{6}
$$

$\theta'_{sun}$ is the sun zenith angle in water, $\theta'_v$ the viewing zenith angle in water, $v_w$ the wind speed in units of (m s$^{-1}$), $K_d$ the diffuse attenuation coefficient of downwelling irradiance, $k_{uW}$ the attenuation coefficient for upwelling radiance originating from the water layer, $k_{uB}$ the attenuation coefficient for upwelling radiance from the bottom, $R^b_{rs}$ the bottom substrate albedo (irradiance reflectance) and $A_{rs,1}$ and $A_{rs,2}$ are empirical coefficients close to one. For the equations of the attenuation coefficients $K_d(\lambda)$, $k_{uW}(\lambda)$ and $k_{uB}(\lambda)$, see [23,24].

The SIOPs are chosen as follows (see Figure 2):

- $a^*_{phy}(\lambda)$ is the specific absorption coefficient of green algae from the database of the software WASI [27]. It is based on an absorption measurement of the green algae *Mougeotia* sp., grown as pure culture in the laboratory [28], which was later fitted for extension to the near infrared and rescaled to 0.023 m$^2$ mg$^{-1}$ at 674 nm to match field measurements from two German lakes [29].
- $a^*_{NAP}(\lambda)$ is approximated by an exponential equation with slope $S_{NAP} = 0.011$ nm$^{-1}$ [30] and the specific absorption coefficient of 0.027 m$^2$ g$^{-1}$ at 440 nm [31].
- $b^*_{b,phy}(\lambda)$ is the specific backscattering coefficient from normal clear water in Lake Garda, dominated by green algae (provided by C. Giardino, personal communication).
- $b^*_{b,NAP}$ (555) = 0.011 m$^2$ g$^{-1}$ and $n = 0.75$ were calculated by averaging measurements from lakes in Italy, Estonia, the Netherlands and Finland using Table 3 of [32].

Some of the simulations of optically shallow waters represent saltwater environments such as seagrasses, macro-algae and coral reefs. Nevertheless, we used mostly freshwater SIOPs. As the optically shallow water simulations were carried out with low concentrations of OACs (scenario Y−: $C$ = 1 mg m$^{-3}$, $X = 1$ g m$^{-3}$, $Y = 0.2$ m$^{-1}$), the effect on the results of not choosing an additional set of saltwater SIOPs is considered to be minimal.

### 2.3. Determination of the Optimal Spectral Resolution

To capture the information content of a reflectance spectrum, a measurement must resolve the spectral features of the spectrum, particularly the peaks, dips and shoulders. These changes in steepness are given by the first derivative $\partial R_{rs}/\partial\lambda$. It can be measured by a real sensor only approximately, depending on the measurement's quantization $\Delta R_{rs}$ and the sensor's spectral resolution $\Delta\lambda$: $\Delta R_{rs}/\Delta\lambda \approx \partial R_{rs}/\partial\lambda$. For given $\Delta R_{rs}$, the ideal spectral resolution is thus:

$$\Delta\lambda = \left|\frac{\Delta R_{rs}}{\partial R_{rs}/\partial\lambda}\right|. \tag{7}$$

At wavelength regions of large reflectance changes, the spectrum must be sampled more frequently than at regions of small gradients, hence $\Delta\lambda$ is inversely related to $\partial R_{rs}/\partial\lambda$. To minimize artifacts introduced by sensor or model noise at spectral regions where the reflectance spectrum is flat or has minima or maxima, $\Delta\lambda = 20$ nm is set for $|\partial R_{rs}/\partial\lambda| < 10^{-6}$ sr$^{-1}$ nm$^{-1}$. Equation (7) defines the optimal spectral resolution for a measurement with a noise-equivalent reflectance of $\Delta R_{rs}$. Equation (8) is used for $\Delta R_{rs}$ because it addresses the resolution of the spectral shape of a measurement.

### 2.4. Determination of the Optimal Radiometric Sensitivity

#### 2.4.1. Rrs. Quantization

The $R_{rs}$ quantization $\Delta R_{rs}$ is defined in this study as the smallest difference of remote sensing reflectance that can be resolved by a measurement. As it is determined by the noise of a measurement, $\Delta R_{rs}$ is also called noise-equivalent remote sensing reflectance difference. It should not be confused with the radiometric resolution of a sensor, which is always in energy units (e.g., photons, radiance). The required resolution $\Delta R_{rs}$ is derived using two approaches. Both approaches are applied to all spectra simulated for all deep water scenarios.

The first approach addresses the measurement of remote sensing reflectance spectra for determining the absolute values of environmental parameters from a spectral analysis of $R_{rs}(\lambda)$. It is based on the postulation that the dynamics of a spectrum $R_{rs}(\lambda)$ should be sampled at a typical resolution of 1%. Hence, the noise-equivalent remote sensing reflectance difference of this approach, $\Delta R_{rs,1}$, is calculated as 1% of the difference between the reflectance maximum, $R_{rs}(\lambda_{max})$, and the reflectance minimum, $R_{rs}(\lambda_{min})$:

$$\Delta R_{rs,1} = 0.01 \left| R_{rs}(\lambda_{max}) - R_{rs}(\lambda_{min}) \right|. \tag{8}$$

The subscript "1" refers to approach number 1. The wavelength interval from 400 to 800 nm is taken to determine the wavelengths $\lambda_{max}$ and $\lambda_{min}$ of maximum and minimum reflectance.

The second approach addresses the measurement of $R_{rs}$ differences for quantifying changes in environmental parameters. It is oriented on the postulation that a measurement should be sensitive to changes in the parameter of interest ($x$) in the order of 10%. It first determines the wavelength $\lambda_{max}$ which is most sensitive to reflectance changes induced by $x$. The remote sensing reflectance difference at $\lambda_{max}$, induced by a 10% change in $x$, is then taken to define the noise-equivalent remote sensing difference:

$$\Delta R_{rs,2} = \left| R_{rs}(\lambda_{max},\ 1.1x) - R_{rs}(\lambda_{max},\ x) \right|. \tag{9}$$

The subscript "2" refers to approach number 2.

### 2.4.2. Absolute Radiometric Resolution

The study simulates measurements in units of remote sensing reflectance ($R_{rs}$), which is independent of light intensity and thus not suitable for assessing the capability of radiance sensors for resolving the spectral shape of $R_{rs}$ or detecting $R_{rs}$ differences induced by changes in optically active environmental parameters. Relative $R_{rs}$ units are converted to absolute radiance ($L$) units by multiplying $R_{rs}$ with the illumination of the target in terms of downwelling irradiance $E_d$:

$$L(\lambda) = R_{rs}(\lambda) \times E_d(\lambda). \tag{10}$$

Similarly, the radiance difference $\Delta L$ induced by a remote sensing reflectance difference of $\Delta R_{rs}$ is given by:

$$\Delta L(\lambda) = \Delta R_{rs}(\lambda) \times E_d(\lambda). \tag{11}$$

To estimate radiometric sensor requirements, the illumination $E_d(\lambda)$ is simulated for sun zenith angles of 0°, 20°, 40°, 60° and 70°, using MODTRAN-6 [33]. Radiance differences $\Delta L(\lambda)$ are calculated using Equation (11) for $\Delta R_{rs}$ values of $10^{-3}$, $10^{-4}$, $10^{-5}$ and $10^{-6}$ sr$^{-1}$.

### 2.4.3. Signal-to-Noise Ratio

The signal-to-noise ratio,

$$\text{SNR}(\lambda) = \frac{L(\lambda)}{NEL(\lambda)} \tag{12}$$

specifies the sensor-induced noise for measuring a radiance spectrum $L(\lambda)$. *NEL* is the radiance corresponding to the radiometric sensitivity of a sensor. It is important to note that *NEL* is a sensor parameter, while SNR is a measurement parameter, as it depends on the measured radiance, i.e., on the illumination and reflectance of the target (Equation (10)). If real measurements are used to compare sensors rather than laboratory-based *NEL* data, the SNRs derived from the measurements have to be converted to a common reference spectrum $L(\lambda)$ [34]. This is however only an approximate sensor comparison because the SNR derived from measurements further depends on photon noise and on the $R_{rs}$ variability of the averaged measurements.

Over water, most of the upwelling radiance at the top of the atmosphere (TOA) originates from the atmosphere, thus the SNR at TOA is governed by the radiance contribution from the atmosphere, called path radiance $L^{path}$. The TOA radiance is related to $L^{path}$ and $R_{rs}$ as follows:

$$L^{TOA}(\lambda) = L^{path}(\lambda) + t_A(\lambda) \times E_d(\lambda) \times R_{rs}(\lambda). \tag{13}$$

$t_A(\lambda)$ is the transmission of the atmosphere for the upwelling radiance and $E_d(\lambda)$ is the downwelling irradiance at the bottom of the atmosphere (BOA). A $R_{rs}$ difference of $\Delta R_{rs}$ induces a radiance difference at TOA of

$$\Delta L^{TOA}(\lambda) = t_A(\lambda) \times E_d(\lambda) \times \Delta R_{rs}(\lambda). \tag{14}$$

To resolve this difference, a sensor on a satellite must have a noise-equivalent radiance of $NEL \leq \Delta L^{TOA}(\lambda)$. For the minimum requirement of $NEL = \Delta L^{TOA}(\lambda)$, a measurement has a signal-to-noise ratio of

$$SNR^{TOA}(\lambda) = \frac{L^{TOA}(\lambda)}{\Delta L^{TOA}(\lambda)} = \frac{L^{path}(\lambda)}{t_A(\lambda) \times E_d(\lambda) \times \Delta R_{rs}(\lambda)} + \frac{R_{rs}}{\Delta R_{rs}}. \tag{15}$$

This means that the SNR at TOA is the sum of two components, the one related to the path radiance and the other to the remote sensing reflectance at BOA:

$$SNR^{TOA}(\lambda) = SNR^{path}(\lambda) + SNR^{BOA}(\lambda) \tag{16}$$

with

$$SNR^{path}(\lambda) = \frac{L^{path}(\lambda)}{t_A(\lambda) \times E_d(\lambda) \times \Delta R_{rs}(\lambda)} \tag{17}$$

and

$$SNR^{BOA}(\lambda) = \frac{R_{rs}}{\Delta R_{rs}}. \tag{18}$$

Equation (17) expresses a measurement requirement for the path radiance: the required $SNR^{path}$ is inversely proportional to the $R_{rs}$ difference to be resolved.

## 3. Results

### 3.1. Spectral Resolution

The spectral resolution of a measurement required for capturing the spectral details of reflectance is calculated using Equation (7). The simulations were made by iterating C, X, Y and S across the ranges specified in Tables 1 and 2 in 50 equidistant steps. For example, the range of the spectral slope of CDOM absorption S is $0.01 - 0.02$ nm$^{-1}$ for all scenarios, hence S was iterated in steps of 0.002 nm$^{-1}$. The simulations for shallow water were made for the 14 bottom substrates shown in Figure 1 and altering water depth in 50 steps from 0.01 to 10 m, i.e., $\Delta R_{rs}$ in Equation (7) represents the induced changes in $R_{rs}$ for depth differences of 20 cm.

For illustration purposes, 2D plots are used, with wavelength on the *x*-axis and the variable parameter on the *y*-axis, and the calculated spectral resolution is color-coded using the color scheme shown in Figure 3.

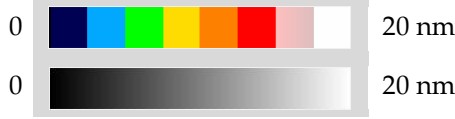

**Figure 3.** Color coding of the spectral resolution plots in Figures 4–6. Top: Scheme used for $\Delta R_{rs} \geq 10^{-6}$ sr$^{-1}$. The colors change in steps of 2.5 nm, i.e., dark blue is 0 to 2.5 nm, light blue 2.5 to 5.0 nm, and so on. Bottom: Scheme used for $\Delta R_{rs} < 10^{-6}$ sr$^{-1}$. The gray values change gradually from black (0 nm) to white ($\geq$20 nm).

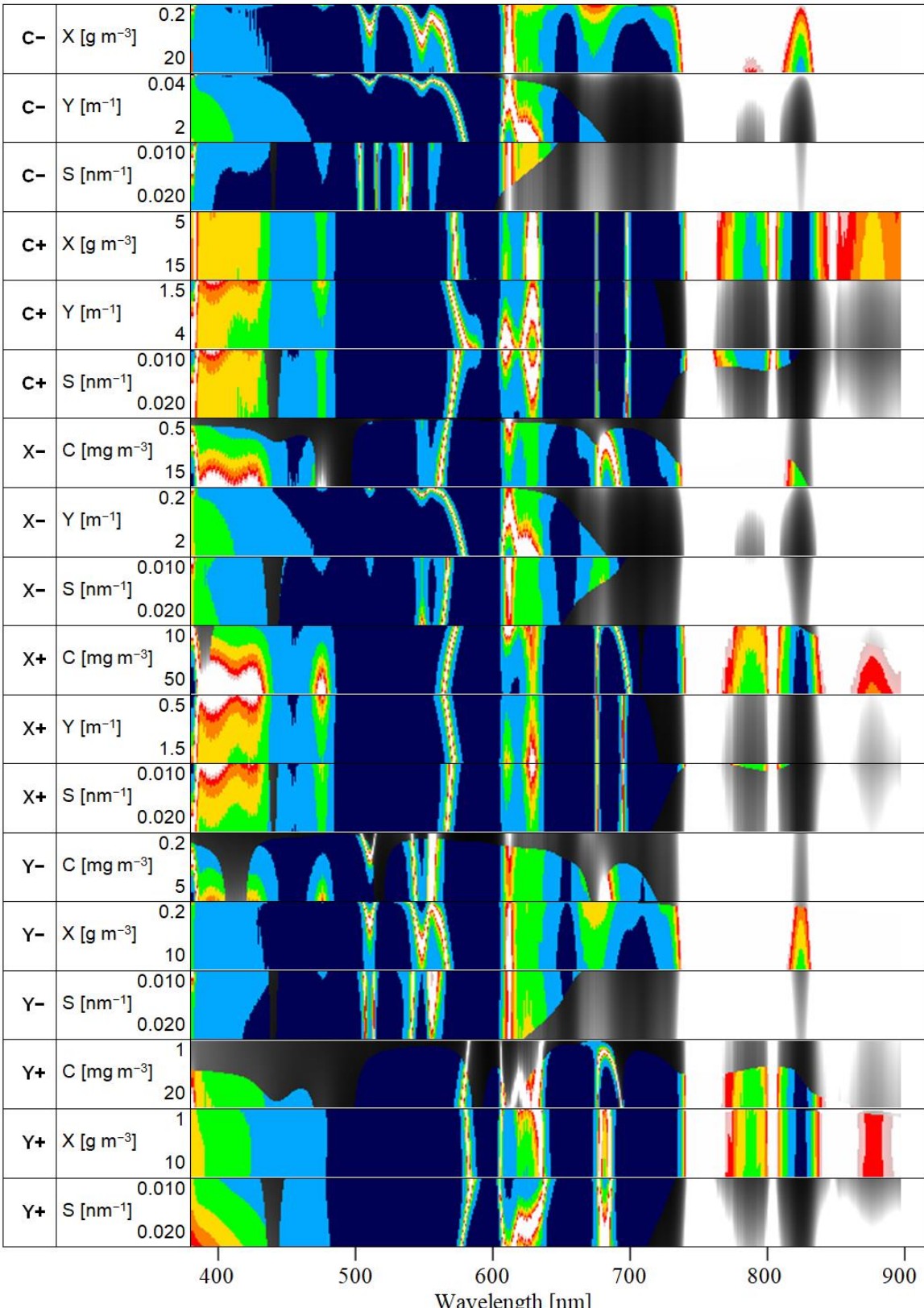

**Figure 4.** Optimal spectral resolution for capturing the spectral details in reflectance measurements for the standard scenarios of optically deep water. For model parameterization, see Table 1; for legend of colors, see Figure 3.

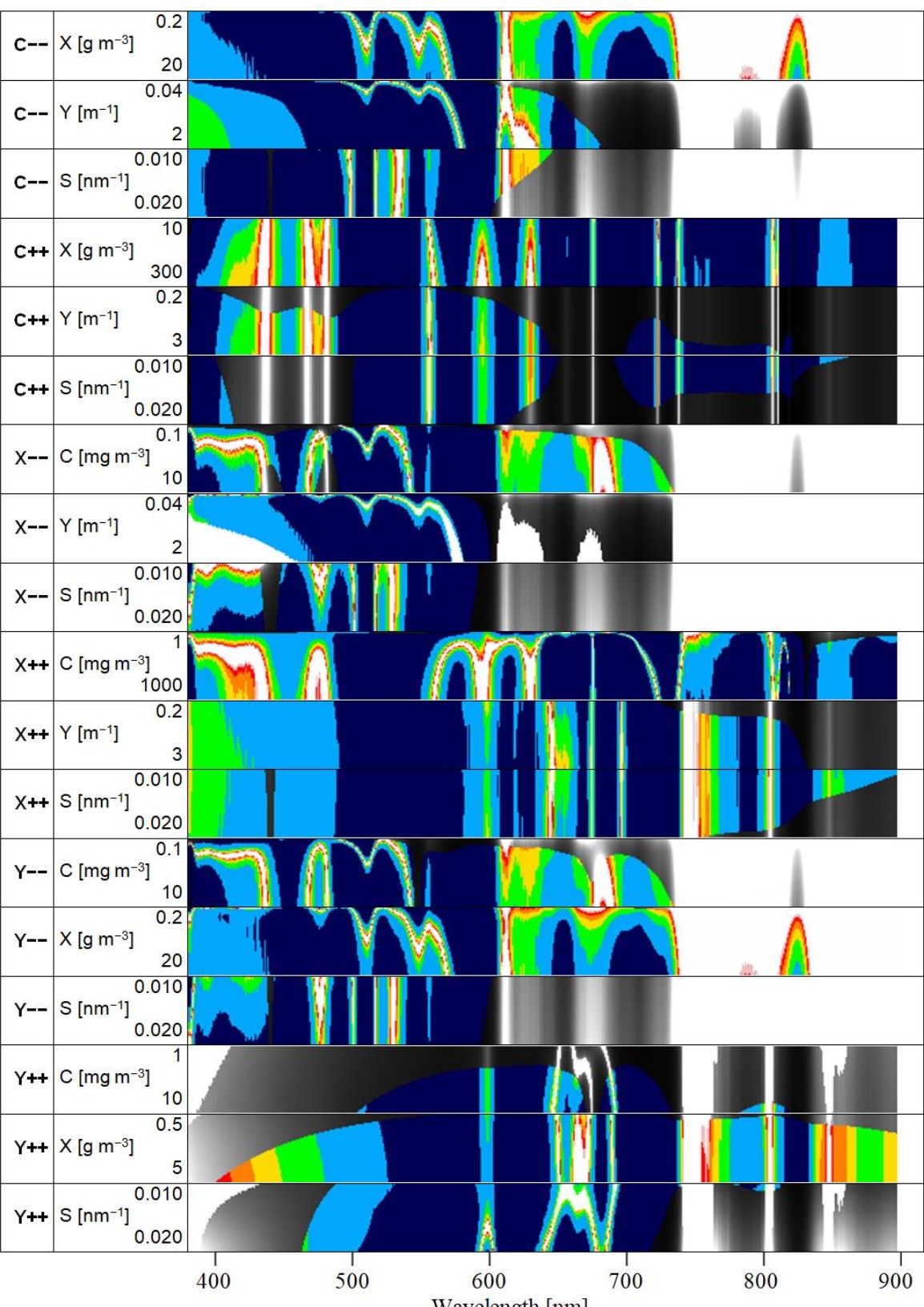

**Figure 5.** Optimal spectral resolution for capturing the spectral details in reflectance measurements for the extreme scenarios of optically deep water. For model parameterization, see Table 2; for legend of colors, see Figure 3.

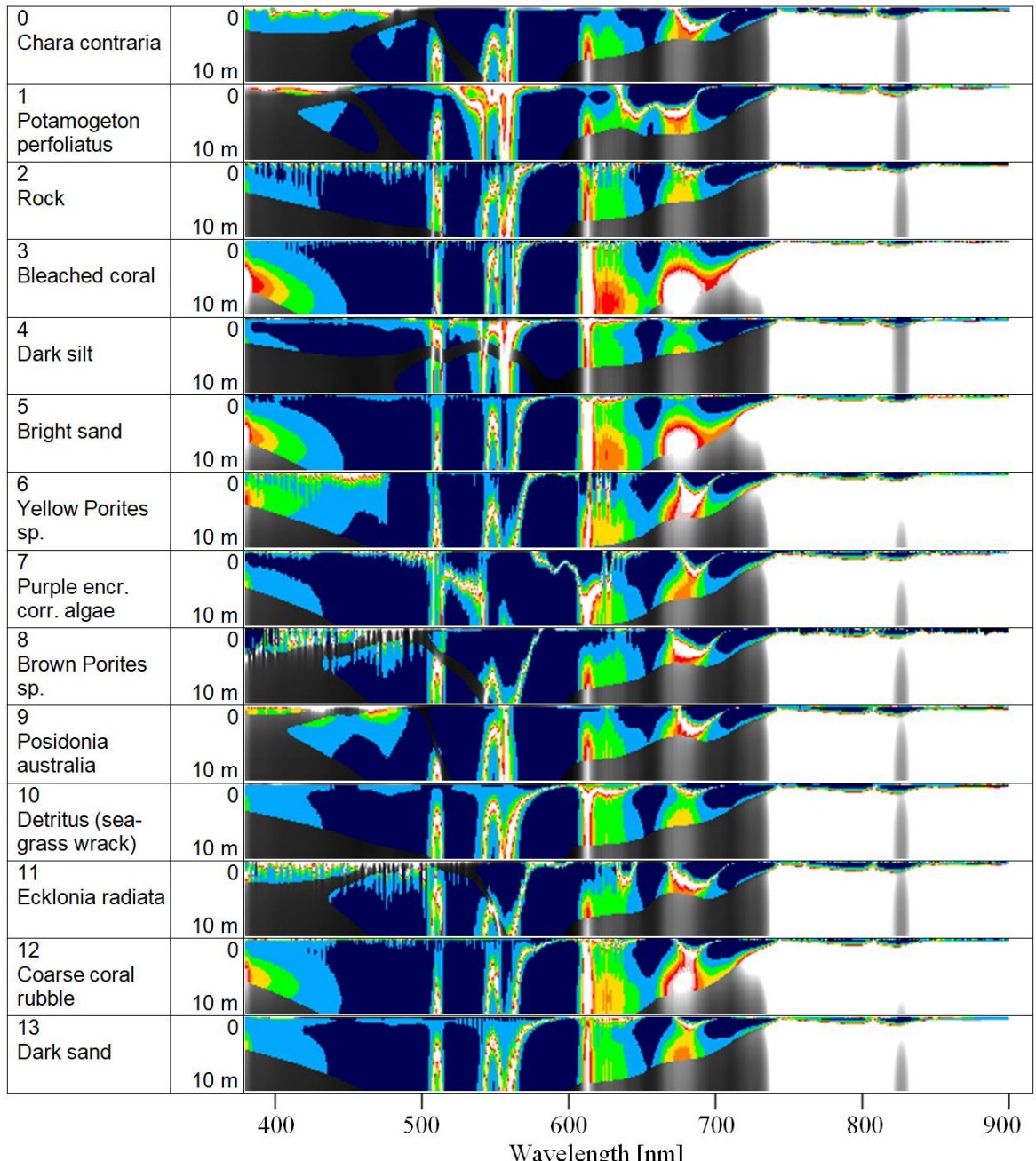

**Figure 6.** Optimal spectral resolution for capturing the spectral details in reflectance measurements for optically shallow waters. For input model parameterization values, see Section 2.2.; for legend of colors, see Figure 3.

The color coding of the 2D plots changes for a radiometric difference of less than $10^{-6}$ sr$^{-1}$ from colors to gray values. This allows us to distinguish between spectral regions in which the induced reflectance changes might be measurable, at least using sensitive field instruments with long integration times (colors), and regions where such differences are below sensor detection limits (gray values).

### 3.1.1. Optically Deep Water

Three 2D-plots were calculated for each deep water scenario. Each plot shows the optimal spectral resolution $\Delta\lambda$ as a function of one parameter covering the scenario-specific range, while the other parameters were kept constant at their scenario-typical values. Figure 4 shows the results for the

standard scenarios and Figure 5 for the extreme scenarios. Summaries of all scenarios are shown below in Figure 6 in terms of medians.

The colored areas in the 2D plots show the wavelengths and concentrations for which 10% changes in the parameters C, X, Y and S induce principally detectable $R_{rs}$ differences above $10^{-6}$ sr$^{-1}$. The different colors represent the associated spectral changes, indicating the lowest useful spectral bandwidths according to the color coding in Figure 3. Spectrally finer resolved measurements cannot capture additional information. Similarly, the areas in gray tones represent wavelengths and concentrations where changes in C, X, Y and S above 10% are required in order to induce detectable $R_{rs}$ differences, if at all, and the gray values represent the lowest useful spectral bandwidths.

Figures 4 and 5 show that the spectral details of $R_{rs}$ are most pronounced in all standard scenarios and most extreme scenarios from about 480 to 600 nm and from 630 to 730 nm. In these regions (dark blue), spectrally highly resolved measurements with resolutions below 2.5 nm can be used to gather spectral information. The spectral details of $R_{rs}$ decrease below 480 nm and in the range of 600 to 630 nm, and they drop markedly above 730 nm. The decrease in the blue (below 480 nm) is more pronounced for scenarios with high concentrations of water constituents (C+, C++, X+, X++, Y+, Y++) than for low concentrations (C−, C−−, X−, Y−). The decrease in the red (600 to 630 nm) is less scenario-specific.

The marked drop in the NIR above 730 nm is caused by the strong increase in pure water absorption. Here, pure water dominates the absorption spectrum of the water body to such an extent that only very high CDOM concentrations (scenarios Y+ and Y++) induce a measurable effect. With the exception of these scenarios, changes in $R_{rs}$ are almost entirely caused by scattering. Since, for most simulated cases, NAP dominates scattering, and scattering by phytoplankton is significantly lower (and CDOM does not scatter at all), measurable effects above 730 nm are mainly related to NAP and to a lesser extent to phytoplankton. The lack of spectral details in $R_{rs}$ makes spectrally highly resolved measurements in the NIR of little use, except for high NAP concentrations at specific spectral regions with local minima of pure water absorption. The most pronounced minimum at 810 nm corresponds to the spectral region used by Kutser et al. [35–37] in black lakes.

### 3.1.2. Optically Shallow Water

The optimal spectral resolution $\Delta\lambda$ was calculated for each optically shallow water scenario using Equation (7) by changing the water depth in steps of 20 cm from 0 to 10 m to simulate the induced changes in $R_{rs}$. Figure 6 shows the resulting 2D plots. Induced $R_{rs}$ differences below $10^{-6}$ sr$^{-1}$ are shaded in gray in order to indicate that measuring these differences is very challenging, even for field instruments, and impossible for current satellite sensors.

The gray shaded areas of Figure 6 illustrate that the spectral interval that can be used for deriving information about the sea floor becomes narrower with increasing water depth. Common to all bottom substrates is the upper boundary, near 740 nm: above this wavelength, light reflected at the bottom is detectable only in very shallow waters of typically less than 1 m depth. Similar conclusions were drawn by Jay et al. [38], who investigated the discrimination of sand, seagrass, brown algae and oyster bags from hyperspectral measurements. They found that at a water depth of 1 m and for most wavebands larger than 700 nm, the water attenuation is already such that the covariance matrix of these four substrates is mainly dominated by environmental noise.

At short wavelengths, the bottom affects $R_{rs}$ only up to a water depth of a few meters for the considered water type of scenario Y−. Only the spectral range from approximately 500 to 600 nm allows the measuring of light reflected from the bottom substrates at 10 m water depth. The width of this interval depends on the bottom albedo (refer for comparison to the corresponding albedo spectra in Figure 1): the darker the substratum, the narrower the range; the brighter the substratum (e.g., bright sand or bleached coral), the wider the spectral range and the greater the depth at which a measurable signal is present.

The optimal spectral resolution $\Delta\lambda$ changes most significantly between 0 and approximately 1 m water depth. The spectral signature of these very shallow waters resembles more land surfaces than water bodies; hence, the simulations for water depths above 1 m are better suited to drawing general conclusions for remote sensing of optically shallow water ecosystems.

The dark blue areas of Figure 6, with values of $\Delta\lambda$ below 2.5 nm, show that for water depths between 1 and 10 m, the spectral range of about 450 to 600 nm bears the most information, which is in accordance with the simulation studies of Hochberg et al. [39] and Kutser et al. [40], who found that coral reef benthos are best discriminated at wavelengths shorter than 580 nm. Up to depths of approximately 5 m, the spectral range between 630 and 730 nm can also contain significant spectral information, depending on the substratum type. Botha et al. [18] have shown with data from the sensors QuickBird, WorldView-2 and CASI that increased spectral resolution leads to more substratum types being discernible from each other at greater depths and enhances bathymetry retrieval. Figure 6 indicates the lower useful limits of spectral bandwidth.

### 3.1.3. Optimal Spectral Resolution

The 2D plots from the previous sections show that the optimal spectral resolution for capturing the spectral details present in reflectance spectra depends on the water type. To obtain globally representative values, the simulations for all water types of the standard scenarios (Figure 4), extreme scenarios (Figure 5) and optically shallow water scenarios (Figure 6) are averaged for all cases in which the induced reflectance differences are $\Delta R_{rs} \geq 10^{-6}$ sr$^{-1}$. To reduce the impact of unusually high $\Delta\lambda$ values, the median is used rather than the mean. Figure 7 shows the results.

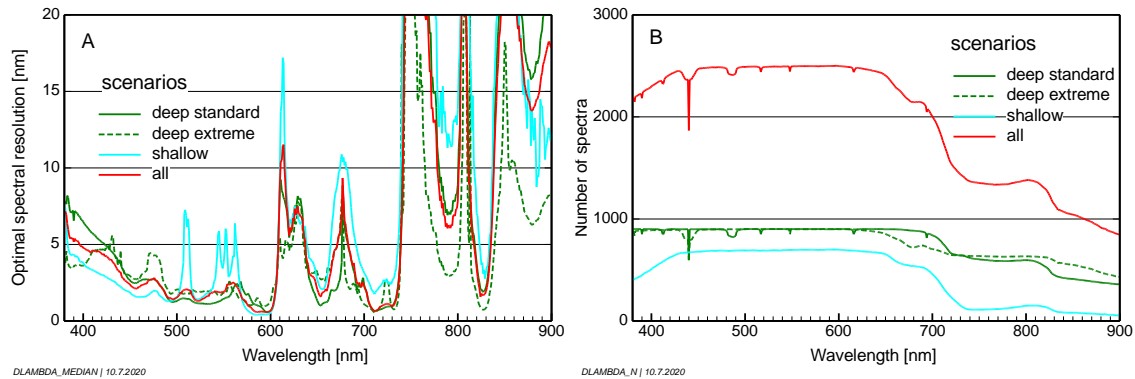

**Figure 7.** (**A**): Medians of the optimal spectral resolutions for capturing the details of $R_{rs}$ spectra. (**B**): Number of spectra used for calculating the medians.

It can be seen in Figure 7A that the optimal spectral resolution depends on wavelength. It ranges from 0.6 to 4.8 nm in the blue-green (400 to 600 nm), from 0.7 to 11.5 nm in the red (610 to 735 nm) and increases significantly at 740 nm, exceeding 20 nm in wide spectral intervals. The averages across all scenarios are 2.9 nm from 400 to 735 nm and 13.8 nm from 740 to 900 nm. The spectral pattern is similar for the standard and extreme scenarios of optically deep water, whilst the optically shallow water scenarios produce more variable values in the 500 to 580 nm range.

The results of Figure 7 are based on 2500 simulated spectra. For calculating the median reflectance, differences $\Delta R_{rs} < 10^{-6}$ sr$^{-1}$ were ignored as these are extremely difficult to measure even with sensitive field instruments. This data selection leads to a wavelength dependency of the used number of spectra (Figure 7B). Figure 7B thus illustrates which spectral regions are difficult to measure due to the low dependency of reflectance on water composition, bottom substratum or depth.

Analogous statistical analyses were made for the 75% and 90% percentiles, which represent 75% and 90% of all simulated spectra (no figure). From 400 to 735 nm, the average 75% percentile is 1.7 nm,

and the average 90% percentile is 1.2 nm. From 740 to 900 nm, the corresponding values are 8.3 nm and 2.3 nm.

While the 2D plots of the previous sections and the medians of Figure 7 quantify the lower useful limits of spectral bandwidth for detecting 10% changes in the parameters C, X, Y, 0.002 nm$^{-1}$, changes in S and 20 cm differences in water depth, they do not allow us to derive the upper limits of bandwidth that are still suitable for determining these parameters. All scenarios represent optically complex waters in which $R_{rs}$ is usually affected by several water constituents or substratum types simultaneously at each wavelength. These waters therefore do not allow us to determine an unknown parameter using a single wavelength. The ambiguity of single band retrieval algorithms is reduced for algorithms based on several bands; the more bands, the less pronounced are the ambiguities. However, even hyperspectral measurements with a contiguous series of narrow bands can suffer from ambiguities [36], depending on measurement noise and number, type and concentration of water constituents [37] and bottom substrates. The accuracy of the retrieved parameters depends on the used algorithm and its strategy to handle ambiguities [38], as well as on the spectral range, spectral resolution and radiometric resolution of the measurements. Figures 4–7 help us to estimate the ideal spectral resolution of measurements and determine the useful spectral range for data analysis in different water types, while the results of Section 3.2 provide information concerning the most sensitive wavelengths and the required radiometric resolution. Which reduced spectral resolutions are still sufficient in practice for accurate parameter retrieval can only be determined for specific retrieval algorithms and given measurement noise. Such analysis is, however, out of scope for this paper.

## 3.2. Radiometric Sensitivity

The radiometric sensitivity requirements of inland and coastal waters are studied using the deep water scenarios, since these encompass the darker targets driving the sensor requirements. Shallow waters are generally much brighter and thus their radiometric sensitivity requirements fall well within the optically deep water requirements. The required $R_{rs}$ quantization is estimated using two approaches, Equations (8) and (9). The first is based on a postulation for sampling the reflectance spectrum, the second for resolving concentration changes at the wavelength of maximum sensitivity.

### 3.2.1. Resolving Spectral Features

The $R_{rs}$ quantization required for sampling the spectral features of a reflectance spectrum is calculated using Equation (8). It specifies the noise-equivalent remote sensing reflectance $\Delta R_{rs,1}$ in terms of 1% of the maximum remote sensing reflectance difference, in the range of 400 to 800 nm.

The simulations were made by keeping C, X or Y constant at the scenario-specific values given in Tables 1 and 2 and iterating the other three parameters in 10 steps in the ranges also given in these tables. In this manner, $10^3 = 1000$ spectra were simulated for each scenario, from which $\Delta R_{rs,1}$ was derived using Equation (8). The results are shown in Figure 8.

Figure 8 shows that the $R_{rs}$ differences generally increase with increasing NAP (X) and decreasing CDOM absorption (Y), but they do not depend much on phytoplankton concentration (C). The reflectance spectra of all considered scenarios can be sampled with a radiometric resolution of 1% or better for $\Delta R_{rs,1} = 10^{-6}$ sr$^{-1}$. $\Delta R_{rs,1} = 10^{-5}$ sr$^{-1}$ is sufficient for most parameter combinations of the standard scenarios, except scenarios Y+ and C− for X < 1 g m$^{-3}$, scenario C− for Y > 0.4 m$^{-1}$ and scenario X− for Y > 2 m$^{-1}$. The extreme scenarios more frequently require $\Delta R_{rs,1}$ between $10^{-5}$ and $10^{-6}$ sr$^{-1}$. The median $R_{rs}$ differences are $1.1 \times 10^{-4}$ sr$^{-1}$/$4.6 \times 10^{-5}$ sr$^{-1}$ for the considered C values of the standard/extreme scenarios (upper row of Figure 8), $1.2 \times 10^{-4}$ sr$^{-1}$/$2.3 \times 10^{-4}$ sr$^{-1}$ for the X values (middle row of Figure 8) and $1.2 \times 10^{-4}$ sr$^{-1}$/$2.6 \times 10^{-4}$ sr$^{-1}$ for the Y values (lower row of Figure 8). The overall median is $1.5 \times 10^{-4}$ sr$^{-1}$.

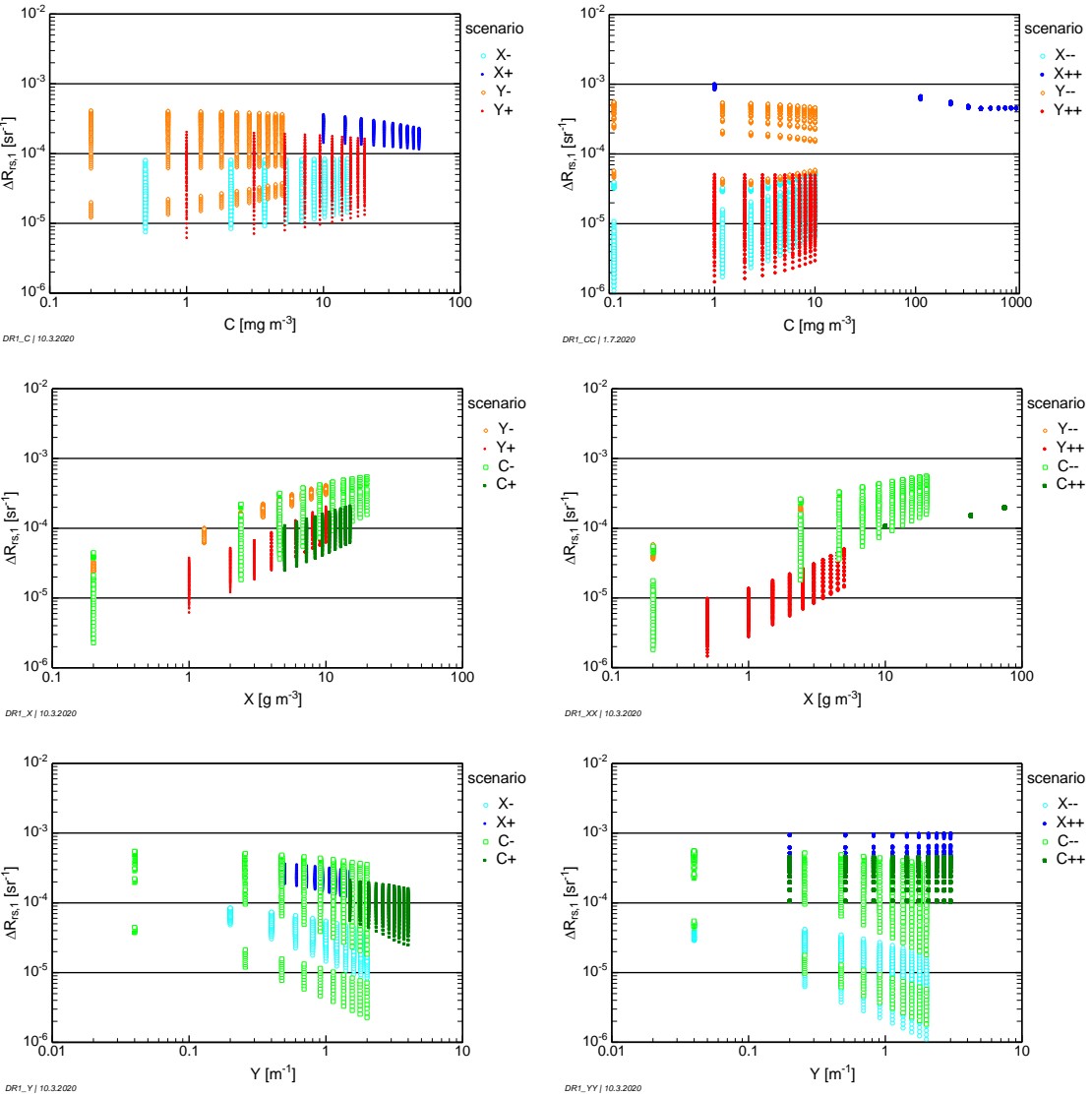

**Figure 8.** $R_{rs}$ differences corresponding to 1% of the dynamic range of $R_{rs}$. Left column: standard scenarios; right column: extreme scenarios of optically deep water.

### 3.2.2. Resolving Concentration Changes

The $R_{rs}$ quantization required for resolving relevant changes of a model parameter is calculated using Equation (9). It specifies the noise-equivalent remote sensing reflectance $\Delta R_{rs,2}$ as the maximum change in $R_{rs}$ in the wavelength range of 400 to 800 nm induced by a 10% change in the parameter of interest. C, X, Y and S were treated as the parameters of interest.

The simulations were made by iterating for each scenario the three variable parameters in 10 steps in the ranges given in Tables 1 and 2. For each parameter combination, $\Delta R_{rs,2}$ was derived using Equation (9). This equation requires a double calculation of the reflectance spectrum, i.e., for values of x and 1.1x for the parameter of interest. For example, the sensitivity analysis for concentration changes of phytoplankton in scenario X− was made by iterating C from 0.5 to 15 mg m$^{-3}$ in 10 steps. In each step, X = 1 g m$^{-3}$ was set, Y was changed from 0.2 to 2 m$^{-1}$ and S was changed from 0.010 to 0.020 nm$^{-1}$. The reflectance spectrum was calculated for C and 1.1C for each parameter combination to obtain the wavelength of maximum sensitivity and the induced reflectance change. The resulting reflectance differences are the 1000 cyan circles labeled X− in diagram "C" of Figure 9.

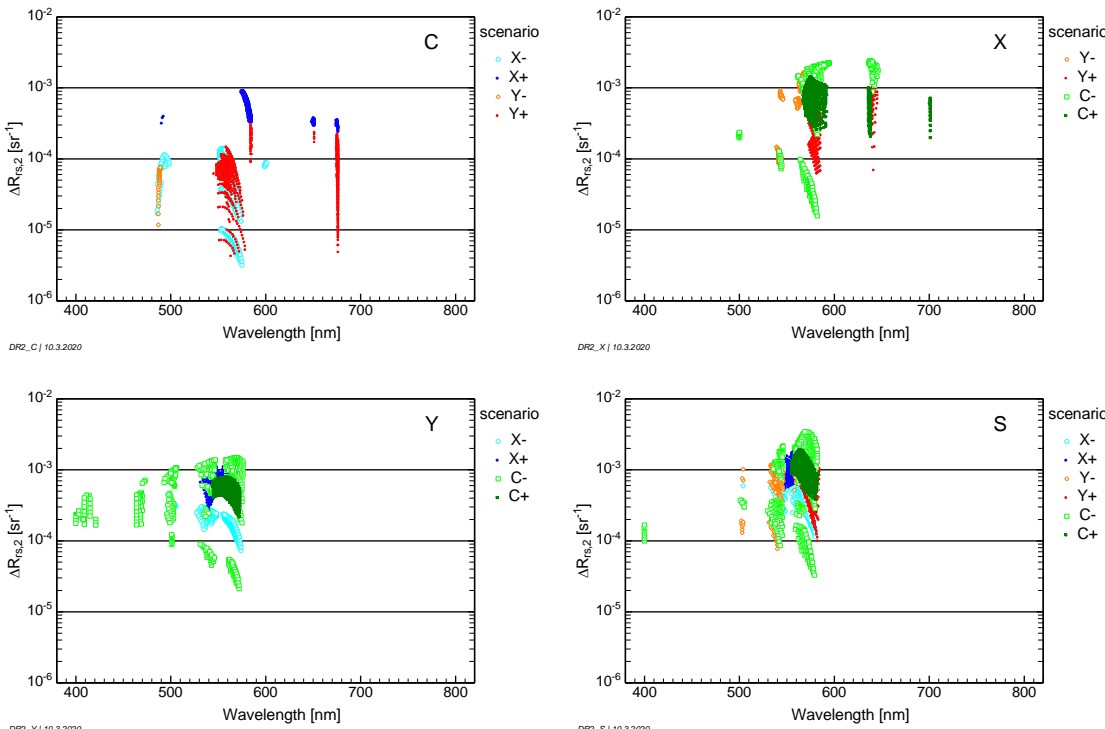

**Figure 9.** Maximum change in $R_{rs}$ for a 10% change in the parameter indicated top right for the standard scenarios. Parameter changes of $f \times 10\%$ alter the shown $\Delta R_{rs}$ values approximately by the factor $f$.

The results of these simulations are shown in Figure 9 for the standard scenarios and in Figure 10 for the extreme scenarios as a function of the wavelength of maximum sensitivity.

Figures 9 and 10 reveal that $\Delta R_{rs,2} = 10^{-5}$ sr$^{-1}$ allows the resolution of 10% changes of X, Y and S for all standard scenarios and for the majority of the conditions studied for the extreme scenarios. However, C requires $\Delta R_{rs,2}$ as fine as to $3 \times 10^{-6}$ sr$^{-1}$ for the standard scenarios and even below $1 \times 10^{-6}$ sr$^{-1}$ for the extreme scenarios. High sensitivity is particularly required for the retrieval of low phytoplankton concentrations in dark waters with low X or high Y (scenarios X−, X−−, Y++).

The derived $\Delta R_{rs,2}$ values are the maximum changes in $R_{rs}$ in the range of 400 to 800 nm. These make 10% changes in the considered parameter principally detectable, but this does not necessarily mean that the parameter can be identified and distinguished from other parameters. Figures 9 and 10 show that C, X, Y and S have no specific spectral region which could be attributed uniquely to one of them, but each can induce strong changes to $R_{rs}$ almost anywhere in the visible range. Thus, in most of the considered water types, classification requires spectral information from other wavelengths with less pronounced $R_{rs}$ changes, i.e., the radiometric resolution should be even higher for quantitative data analysis.

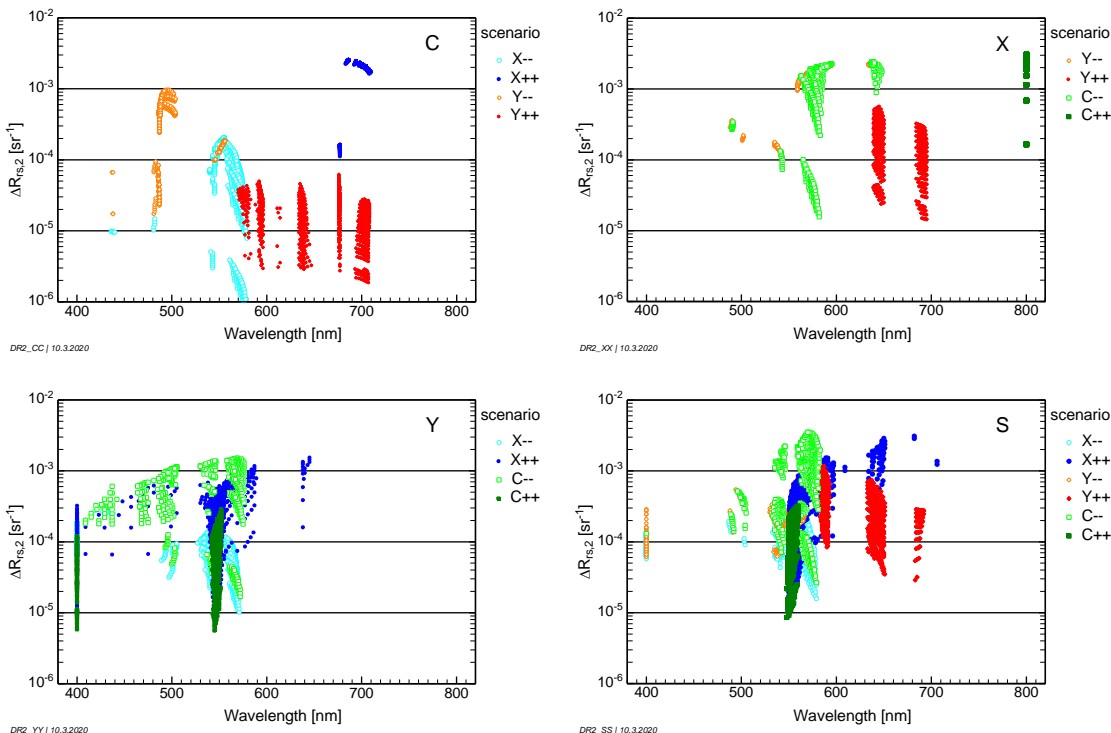

**Figure 10.** Maximum change in $R_{rs}$ for a 10% change in the parameter indicated top right for the extreme scenarios. Parameter changes of $f \times 10\%$ alter the shown $\Delta R_{rs}$ values approximately by the factor $f$.

### 3.2.3. Optimal $R_{rs}$ Quantization

A statistics of the maximum changes in $R_{rs}$ induced by 10% concentration changes and by 0.002 nm$^{-1}$ differences in the spectral slope of CDOM absorption is provided in Figure 11. The left plot shows for each wavelength the medians of $\Delta R_{rs,2}$ across all simulations of Figures 9 and 10. The right plot shows a histogram of the wavelengths which are most sensitive to changes in C, X, Y and S. The wavelengths of the local maxima are labeled.

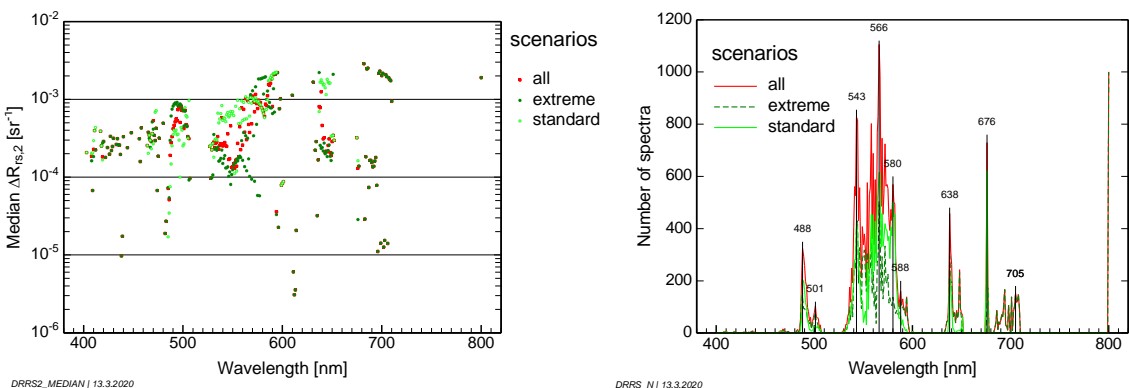

**Figure 11.** Left: Medians of the maximum changes in $R_{rs}$ induced by 10% concentration changes. Right: Number of spectra used for calculating the medians and wavelengths of local maxima.

Most sensitive to changes in C, X, Y and S is the spectral region between 535 and 595 nm. Further prominent ranges are 485–505 nm, 630–650 nm and 675–710 nm. The median of the maximum changes is for most simulated cases between $10^{-4}$ and $10^{-3}$ sr$^{-1}$. The median of $1.5 \times 10^{-4}$ sr$^{-1}$ obtained in

Section 3.2.1 for resolving the spectral shape of reflectance spectra is inside this range, i.e., the two approaches for estimating the optimal radiometric sensitivity lead to comparable results.

### 3.2.4. Radiometric Sensor Requirements

To estimate radiometric sensor requirements, radiance differences $\Delta L(\lambda)$ were simulated for $\Delta R_{rs}$ values of $10^{-3}$, $10^{-4}$, $10^{-5}$ and $10^{-6}$ sr$^{-1}$, which represent the four lowermost horizontal lines of Figure 8 to Figure 10 and the left plot of Figure 11. The calculations were made for sun zenith angles of 0°, 20°, 40°, 60° and 70°, using the mid-latitude summer atmospheric model of Modtran-6 [33] for a horizontal visibility of 50 km. Figure 12 shows the resulting radiance differences. To make them comparable with the sensitivity of existing satellite sensors, the noise-equivalent radiances *NEL* of MODIS and OLCI on Sentinel-3 are also shown, representing instruments optimized for ocean color remote sensing. Sentinel-2 has also been included as an example of a state-of-the art land sensor. *NEL* is the radiance corresponding to the radiometric sensitivity of a sensor.

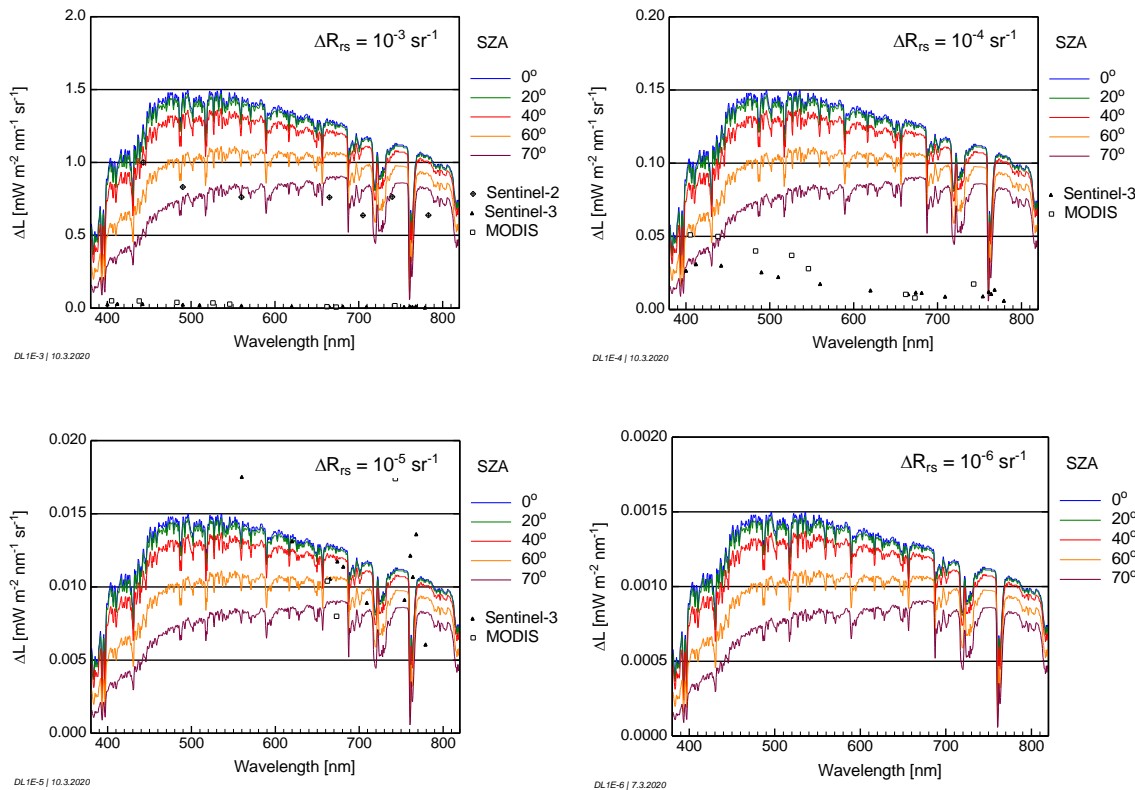

**Figure 12.** Radiance differences induced by $R_{rs}$ changes of $10^{-3}$, $10^{-4}$, $10^{-5}$, $10^{-6}$ sr$^{-1}$ for clear atmospheric conditions and sun zenith angles of 0°, 20°, 40°, 60°, 70°; noise-equivalent radiances of Sentinel-2, Sentinel-3 OLCI and MODIS for comparison.

Figure 12 reveals that instruments with sensitivities comparable to Sentinel-2 allow the differentiation of only reflectance differences in the order of $\Delta R_{rs} = 10^{-3}$ sr$^{-1}$ for all considered sun zenith angles, while sensors like OLCI on Sentinel-3 and MODIS can easily measure reflectance differences of $\Delta R_{rs} = 10^{-4}$ sr$^{-1}$, and $\Delta R_{rs} = 10^{-5}$ sr$^{-1}$ can be resolved only above 650 nm. Differences of $\Delta R_{rs} = 10^{-6}$ sr$^{-1}$ are too small for any of these sensors.

### 3.2.5. Signal-to-Noise Requirements

The signal-to-noise ratio at the top of the atmosphere is the sum of a component from the atmosphere, $SNR^{path}$, and another from the water, $SNR^{BOA}$ (Equation (16)). The required $SNR^{path}$ for resolving $R_{rs}$ differences of $10^{-3}$, $10^{-4}$, $10^{-5}$ and $10^{-6}$ sr$^{-1}$ was simulated using Equation (17). These

$\Delta R_{rs}$ values represent the four lowermost horizontal lines of Figure 8 to Figure 10 and the left plot of Figure 11. Modtran-6 [33] was used for calculating $L(\lambda)$ and $E_d(\lambda)$ for a mid-latitude summer atmosphere at a horizontal visibility of 50 km and sun zenith angles of 0°, 20°, 40°, 60° and 70°. Figure 13 shows the results. $SNR^{path}$ increases strongly from long to short wavelengths and can be very large, particularly for $\Delta R_{rs} < 10^{-4}$ sr$^{-1}$. Figure 13 can be useful for relating the environmental parameter $\Delta R_{rs}$ to the measurement parameter SNR at the top of the atmosphere for given $SNR^{BOA}$.

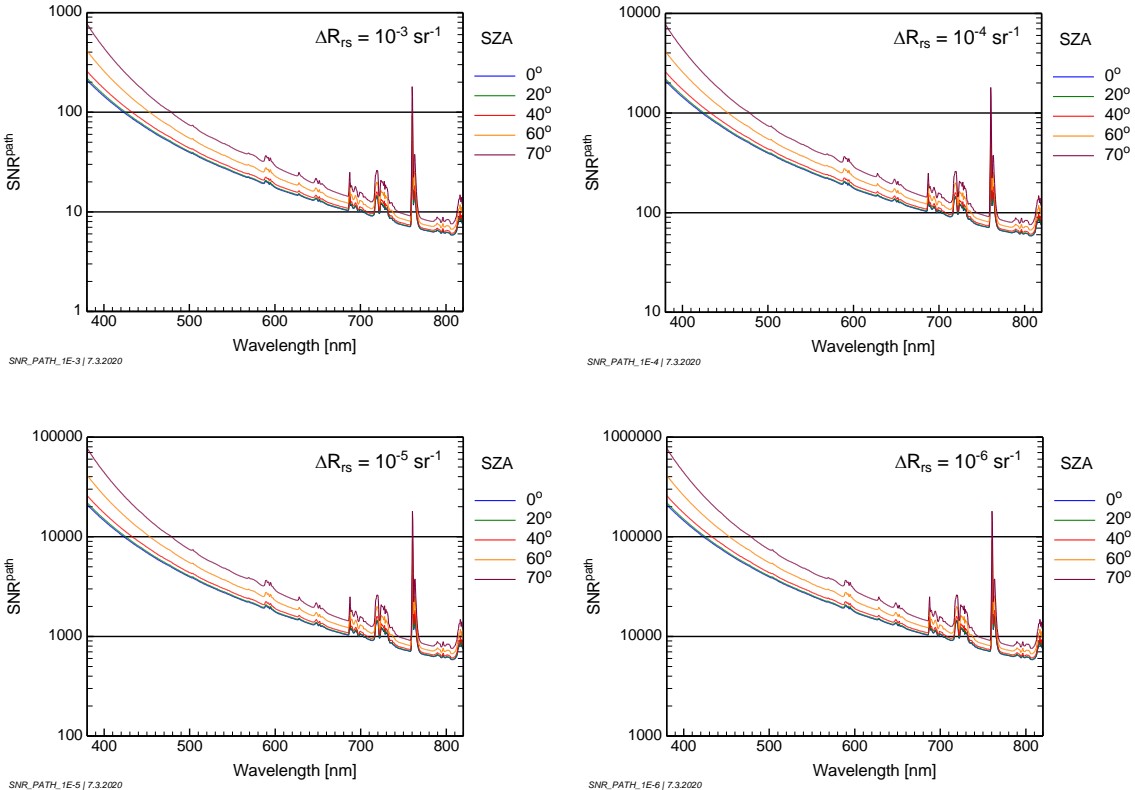

**Figure 13.** Required signal-to-noise ratio of the path radiance for resolving $R_{rs}$ differences of $10^{-3}$, $10^{-4}$, $10^{-5}$ and $10^{-6}$ sr$^{-1}$ under clear atmospheric conditions for sun zenith angles of 0°, 20°, 40°, 60° and 70°.

$SNR^{BOA}$ is determined for both approaches to estimating the required resolution $\Delta R_{rs}$, i.e., capturing the spectral shape of reflectance measurements (Equation (8)) and resolving concentration changes (Equation (9)). The simulated values of $\Delta R_{rs}$ in the first approach have been shown in Figure 8, those of the second approach in Figure 9 to Figure 11.

A summary of the second approach is given in Table 4 for the most sensitive wavelengths. It quantifies $SNR^{BOA}$ and $SNR^{path}$ and their sum, $SNR^{TOA}$, for a sun zenith angle of 40°. The chosen wavelengths represent the labeled maxima of Figure 11 (right). The reflectance differences listed in the $\Delta R_{rs}$ column are taken from Figure 11 (left). They represent the median of the maximum $R_{rs}$ differences over all standard and extreme deep water scenarios induced by 10% concentration changes in water constituents and 0.002 nm$^{-1}$ changes in the spectral slope of CDOM absorption. $\Delta\lambda$ indicates the median optimal spectral resolutions of all scenarios (red curve of Figure 7).

**Table 4.** Wavelengths most sensitive to concentration changes, their medians of the optimal spectral resolution $\Delta\lambda$, their median $\Delta R_{rs}$ of the induced $R_{rs}$ differences and their medians of the signal-to-noise ratio at the water surface ($SNR^{BOA}$), for the atmospheric path radiance at a sun zenith angle of $40°$ ($SNR^{path}$) and at the top of the atmosphere ($SNR^{TOA}$).

| Wavelength (nm) | $\Delta\lambda$ (nm) | $\Delta R_{rs}$ (sr$^{-1}$) | $SNR^{BOA}$ | $SNR^{path}$ | $SNR^{TOA}$ |
|---|---|---|---|---|---|
| 488 | 1.8 | $3.3 \times 10^{-4}$ | 62 | 151 | 213 |
| 501 | 1.6 | $4.0 \times 10^{-4}$ | 21 | 108 | 129 |
| 543 | 1.9 | $2.5 \times 10^{-4}$ | 26 | 118 | 144 |
| 566 | 2.1 | $6.8 \times 10^{-4}$ | 13 | 36 | 49 |
| 580 | 0.8 | $8.0 \times 10^{-4}$ | 16 | 28 | 44 |
| 588 | 0.6 | $2.0 \times 10^{-3}$ | 14 | 11 | 25 |
| 638 | 4.3 | $6.8 \times 10^{-4}$ | 13 | 21 | 34 |
| 676 | 6.7 | $5.6 \times 10^{-5}$ | 56 | 203 | 259 |
| 705 | 1.2 | $1.9 \times 10^{-3}$ | 19 | 5 | 24 |

The values of $\Delta R_{rs}$ and SNR in Table 4 can be interpreted as the lowest useful limits for designing a sensor as they are derived for the one wavelength in the range of 400 to 800 nm with a maximally induced reflectance change, and the median only represents half of the simulations. Most algorithms however also require measurements at wavelengths of lower sensitivities in order to disentangle the influence of different water constituents and eventually bottom substrates, i.e., they analyze the spectral shape of the spectrum.

The values of $\Delta R_{rs}$ required for capturing the spectral shape have been shown in Figure 8 for all simulated cases. Figure 14A summarizes these values in terms of quantiles representing 50%, 75% and 90% of all simulated spectra with $\Delta R_{rs} \geq 10^{-6}$ sr$^{-1}$. The 50% quantile represents the median.

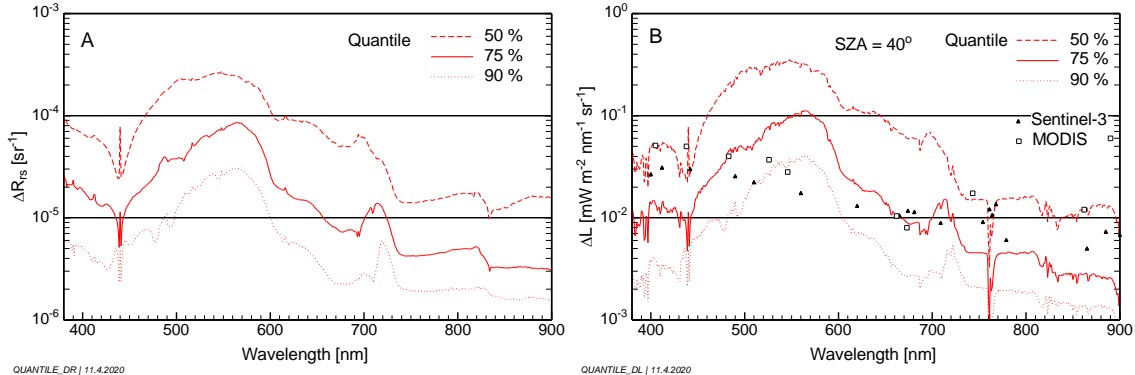

**Figure 14.** Reflectance differences (**A**) and radiance differences (**B**) that must be resolved in order to capture the spectral shape of reflectance spectra with a quantization of 100 levels for 50%, 75% and 90% of all simulated cases. The radiance differences are shown for a sun zenith angle of $40°$ and compared with the noise-equivalent radiances of the sensors OLCI on Sentinel-3 and MODIS.

Figure 14A reveals significant spectral variability of $\Delta R_{rs}$, with very low values above 740 nm. From 400 to 735 nm, the average 50% percentile is $1.2 \times 10^{-4}$ sr$^{-1}$, the average 75% percentile is $2.9 \times 10^{-5}$ sr$^{-1}$ and the average 90% percentile is $1.0 \times 10^{-5}$ sr$^{-1}$. From 740 to 900 nm, the corresponding values are $1.5 \times 10^{-5}$ sr$^{-1}$, $4.0 \times 10^{-6}$ sr$^{-1}$ and $1.8 \times 10^{-6}$ sr$^{-1}$.

Figure 14B shows the corresponding radiance differences $\Delta L$ for a sun zenith angle of $40°$, as obtained using Equation (11). For comparison, the noise-equivalent radiances of the OLCI sensor on Sentinel-3 and the MODIS sensor are also shown. It can be concluded from Figure 14B that satellite sensors with a radiometric sensitivity comparable to OLCI on Sentinel-3 and MODIS are capable of capturing the radiometric details for ~50% of the studied scenarios in the blue (400–450 nm), for

~75–95% in the green (490–550 nm), for ~75% in the red (600–700 nm) and for ~50–60% in the near infrared (720–900 nm).

Combining $\Delta R_{rs}$ from Figure 14A with the corresponding remote sensing reflectance $R_{rs}$ yields $SNR^{BOA}$ according to Equation (18), and applying Equation (17) to $\Delta R_{rs}$ yields $SNR^{path}$. The sum of $SNR^{BOA}$ and $SNR^{path}$ gives $SNR^{TOA}$. Figure 15 shows the quantiles of $SNR^{BOA}$ and $SNR^{TOA}$, which represent 50%, 75% and 90% of all simulated reflectance spectra with $\Delta R_{rs} \geq 10^{-6}$ sr$^{-1}$.

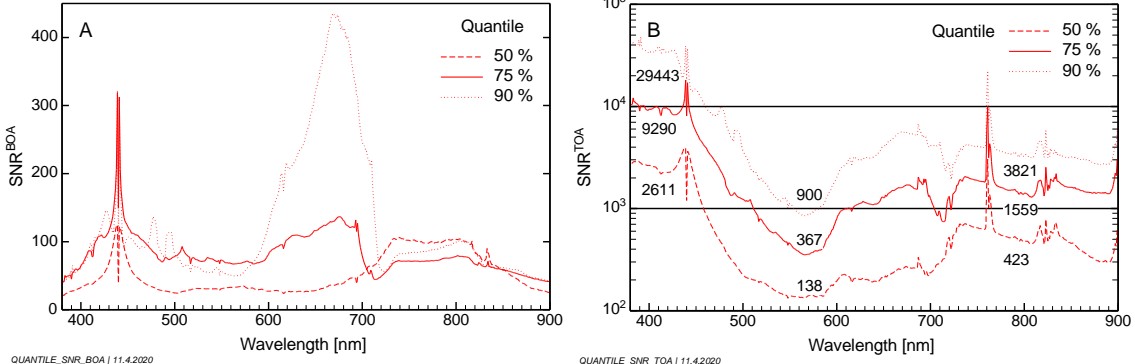

**Figure 15.** Signal-to-noise ratio at the water surface (**A**) and at the top of the atmosphere (B) required for capturing the spectral shape of reflectance spectra with a quantization of 100 levels for 50%, 75% and 90% of all simulated cases. The simulations are shown for a sun zenith angle of 40°. The numbers in (**B**) are the averages for the spectral ranges 400–450 nm, 560–580 nm and 600–900 nm.

The postulation behind the simulations for resolving spectral features is that the maximum difference of a reflectance spectrum from 400 to 800 nm should have a quantization of 100 levels (Equation (8)). Figure 15A translates this postulation into measurement requirements concerning the signal-to-noise ratio at the water surface. The average $SNR^{BOA}$ across all scenarios for the spectral range of 400 to 800 nm is 52 for the 50% quantile (median), 90 for the 75% quantile and 143 for the 90% quantile.

The SNR at the top of the atmosphere is governed by the path radiance, as illustrated above in Figure 13. Figure 15B shows that $SNR^{TOA}$ has a minimum of 560–580 nm, some variability around an adumbrated plateau from 600 to 900 nm and increases rapidly below 560 nm to high values from 400 to 450 nm. The averages in these ranges are given in the figure.

## 4. Summary and Conclusions

The main goal of the paper was to derive globally applicable requirements for measuring remote sensing reflectance ($R_{rs}$) in terms of quantization ($\Delta R_{rs}$) and spectral resolution ($\Delta$). A number of scenarios for optically deep and optically shallow waters were defined, which are expected to cover most of the variety of reflectance spectra of inland and shallow coastal and coral reef waters on Earth. Each optically deep water scenario is represented by the range and typical values of four parameters characterizing the water constituents, while each optically shallow water scenario is defined by the albedo of the bottom substrate or substratum cover type and by a set of water depths. Thousands of reflectance spectra were simulated for each scenario by iterating the scenario-specific parameters. A method was developed (Equation (7)) for deriving the spectral resolution, which captures all spectral details present in $R_{rs}$ for a given quantization $\Delta R_{rs}$. $\Delta R_{rs}$ was derived using two approaches: (1) the spectral shape of $R_{rs}$ should be resolvable with a quantization of 100 levels from 400 to 800 nm; (2) concentration changes in water constituents of 10%, spectral slope differences of 0.002 nm$^{-1}$ of CDOM absorption and depth differences of 20 cm shall produce measurable reflectance differences for at least one wavelength. The results of $\Delta R_{rs}$ and $\Delta \lambda$ are presented in much detail in Sections 3.1 and 3.2. Both parameters change significantly at around 740 nm. From 400 to 735 nm, the medians are $\Delta R_{rs} = 1.2 \times 10^{-4}$ sr$^{-1}$ and $\Delta \lambda = 2.9$ nm; the 90% percentiles are $\Delta R_{rs} = 1.0 \times 10^{-5}$ sr$^{-1}$ and $\Delta \lambda = 1.2$ nm. From 740 to

900 nm, the corresponding values are $\Delta R_{rs} = 1.5 \times 10^{-5}$ sr$^{-1}$ and $\Delta\lambda = 13.8$ nm for the medians and $\Delta R_{rs} = 1.8 \times 10^{-6}$ sr$^{-1}$ and $\Delta\lambda = 2.3$ nm for the 90% percentiles.

The secondary goal was to translate the $\Delta R_{rs}$ results to sensor and measurement requirements for radiance sensors. Radiometric sensor requirements are defined by the noise-equivalent radiance NEL. Radiance differences $\Delta L$, induced by reflectance differences of $\Delta R_{rs}$, are used as proxies for NEL. Simulations of $\Delta L$ were made for specific values of $\Delta R_{rs}$ as a function of the sun zenith angle for a mid-latitude summer atmosphere with 50 km visibility (Figure 12). A statistical evaluation for all scenarios and a sun zenith angle of 40° is presented in Figure 14. To make the results comparable with existing sensors, the derived requirements were compared with the specifications of OLCI on Sentinel-3 and MODIS. These sensors would be capable of capturing the radiometric details for ~50% of the studied scenarios in the blue, ~75–95% in the green, ~75% in the red and ~50–60% in the near infrared for the studied environmental conditions.

Radiometric measurement requirements are defined by the signal-to-noise ratio (SNR). Over water, the SNR at the top of the atmosphere is governed by the path radiance ($L^{path}$). Equations were derived for separating the contributions from the ground and the atmosphere (Equation (16)) and for expressing SNR in terms of $R_{rs}$, $\Delta R_{rs}$ and $L^{path}$ (Equations (17) and (18)). Simulations of typical atmospheric SNRs were made for specific values of $\Delta R_{rs}$ as a function of the sun zenith angle (Figure 13). A statistical evaluation of all scenarios is presented in Figure 15 for a sun zenith angle of 40°. It should be noted that the SNR is a popular parameter for comparing measurements but cannot be used directly for assessing a sensor because the SNR is not a sensor parameter.

Quantitative values of remote sensing reflectance differences ($\Delta R_{rs}$) and optimal spectral resolution ($\Delta\lambda$) were derived for tens of thousands of simulations, which are expected to cover a wide range of inland, coastal and reef waters on Earth. The simulations are based on the assumption that useful remote sensing reflectance ($R_{rs}$) measurements shall be of very high quality in the sense that the spectral shape of $R_{rs}$ should have a quantization of 100 levels, and the spectral resolution should allow the capture of all spectral details which are theoretically resolvable with that quantization. Measurements reaching the derived optimal values $\Delta R_{rs}$ and $\Delta\lambda$ hence represent a best case for subsequent data analysis but a worst case for sensor design.

The CEOS study [1] investigated the technical constraints for an aquatic ecosystem Earth observation system and compared the SNRs at the top of the atmosphere resulting from the desired $\Delta R_{rs}$ values with realistically attainable SNRs for a sensor with a lens aperture of 30 cm and a ground sampling distance between 17 and 33 m in a 400 km orbit. Not surprisingly, they differ considerably, particularly for low sun elevation, turbid atmosphere and for retrieving phytoplankton in dark waters. It can be expected that even sensors at the technological front-end (highly sensitive and low-noise detectors with high dynamic range and large aperture of the telescope) are not able to measure data with such high quality under all circumstances. Hence, trade-offs between spatial, spectral and radiometric resolutions will be necessary to design a satellite sensor for inland and coastal waters. Simulations similar to those presented here can help to optimize these trade-offs. Such simulations should cover, in addition to representative aquatic scenarios like in this study, a reasonable range of atmospheric conditions and sun zenith angles.

To assess the capabilities of a specific sensor for deriving a parameter of interest ($x$), the sequence of steps must be reversed. The input are the sensor parameters (NEL, center wavelength and spectral response of each band, viewing angle), atmospheric conditions and the sun zenith angle. After the atmospheric transmission and the downwelling irradiance at the Earth's surface have been calculated, $\Delta R_{rs}$ can be determined for each wavelength using Equation (14). To convert $\Delta R_{rs}(\lambda)$ into resolvable differences of $x$, partial derivatives must be calculated numerically or an iterative approach applied, since $x$ and $\Delta x$ cannot be expressed explicitly as a function of $R_{rs}$. As $R_{rs}$ measurements in optically complex waters are frequently ambiguous [36], the number and accuracy of retrievable parameters depends on the used algorithm and its strategy for handling ambiguities [38]. The assessment of measurement capabilities should therefore also include retrieval algorithms.

**Supplementary Materials:** The following are available online at http://www.mdpi.com/2072-4292/12/14/2247/s1: simulated data, meta-analyses (statistical analyses, SNR calculations), source code and executable and configuration files of the special version 5.2 of the software WASI, which was used for the simulations.

**Author Contributions:** Conceptualization, P.G. and A.G.D.; methodology, P.G.; software, P.G.; investigation, A.G.D. and P.G.; simulations, P.G.; writing—original draft preparation, P.G.; writing—review and editing, P.G. and A.G.D.; visualization, P.G. All authors have read and agreed to the published version of the manuscript.

**Funding:** This research received no external funding.

**Acknowledgments:** The spectral measurements of the bottom substrates were provided by N. Pinnel, E. Botha and D. Rogge. The specific backscattering coefficient of phytoplankton was provided by C. Giardino.

**Conflicts of Interest:** The authors declare no conflict of interest.

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
