# Peer review of "Spectral and Radiometric Measurement Requirements for Inland, Coastal and Reef Waters"

_remotesensing, doi:10.3390/rs12142247_

Round 1

Reviewer 1 Report

This is a well-written manuscript and timely contribution to specialized sensor design. Attached pdf file contains minor comment.

Author Response

Reviewer #1 suggested three improvements to the wording of the first paragraph of the introduction (lines 34, 35, 38). All three proposed modifications were accepted. He further recommended an overview of section 2 (comment on line 100). A paragraph has been added to provide this overview.

Reviewer 2 Report

I have no comments to provide to the Authors. the entire manuscript is well written, documented, structured and referenced. Discussion is consistent, summary and conclusions are sound and consistent. Figures all are high quality.

Author Response

Reviewer #2 had no recommendations. The authors are grateful to the reviewer for spending his/her valuable time reading and commenting on the manuscript. 

Reviewer 3 Report

The manuscript is interesting, clear and well written. It only needs a final check, e.g.on  page 2. lines 94 - 95 the word "through" is repeated two times. On page 470, Figure 8 DRI_CC | 10.3.2020 should be improved as data overlap on y-axis, unless the Authors did it on purpose.

Author Response

Reviewer #3 discovered a slightly unclean formulation in lines 94-95. The sentence was expressed more clearly. He/she further recommended improving Figure 8 DRI_CC on line 470 as data overlapped on the y-axis. The Figure has been replaced by a Figure in which the x-range has been slightly enlarged so that the overlap disappears.